# Wastewater-Fertigated Short-Rotation Coppice, a Combined Scheme of Wastewater Treatment and Biomass Production: A State-of-the-Art Review

Mirko Hänel [1,2,3,*], Darja Istenič [4,5], Hans Brix [2,3] and Carlos A. Arias [2,3]

1   Technology Transfer Center (TTZ) Bremerhaven, Hausburgstrasse 17, 10249 Berlin, Germany
2   Department of Biology, Aquatic Biology, Aarhus University, Ole Worms Allé 1, 8000 Aarhus, Denmark; hans.brix@bio.au.dk (H.B.); carlos.arias@bio.au.dk (C.A.A.)
3   Centre for Water Technology WATEC, Aarhus University, Vejlsøvej 25, 8600 Silkeborg, Denmark
4   Faculty of Health Sciences, University of Ljubljana, Zdravstvena Pot 5, 1000 Ljubljana, Slovenia; darja.istenic@fgg.uni-lj.si
5   Faculty of Civil and Geodetic Engineering, University of Ljubljana, Jamova Cesta 2, 1000 Ljubljana, Slovenia
*   Correspondence: mhaenel@ttz-bremerhaven.de; Tel.: +49-(0)-471-80-934-191

**Abstract:** Vegetated filters based on short-rotation coppice (SRC) can be used to treat various industrial and municipal wastewater while producing valuable biomass in an economical and sustainable way, showing potential in the field of pollution control and bio-based circular economy. This study provides an overview of the state of the art in wastewater-fertigated SRC systems (wfSRCs) worldwide. Different designs, wastewater sources, tree species and varieties, planting schemes, geographic locations, and climates for wfSRC implementation were identified after conducting a literature review. The performance review includes standard water quality parameters, $BOD_5$, COD, nitrogen, phosphorous, and potassium, as well as the extent of pathogen and emergent contaminant removal and biomass production rates. Identified knowledge gaps and important factors to support the practical implementation of wfSRCs are highlighted. Europe leads the research of wfSRC, followed by North America and Australia. The available publications are mainly from developed countries (73%). The most applied and studied tree species in wfSRC systems are willows (32%), followed by eucalyptus (21%) and poplars (18%). Most of the reviewed studies used domestic wastewater (85%), followed by industrial wastewater (8%) and landfill leachate (7%). Most data show high $BOD_5$ and COD removal efficiencies (80%). There are large differences in the documented total nitrogen and total phosphorus removal efficiencies (12%–99% and 40%–80%, respectively). Enhanced biomass growth in wfSRC systems due to wastewater fertigation was reported in all reviewed studies, and biomass production varied from 3.7 to 40 t DM/ha/yr. WfSRCs seem to have high potential as viable and cost-effective wastewater treatment alternatives to conventional treatment technologies.

**Keywords:** wastewater-fertigated/irrigated short-rotation coppice/plantation (SRC, SRP); wetland treatment; nature-based wastewater treatment systems; energy crops; biomass production systems

## 1. Introduction

### 1.1. Background and Definition of Short-Rotation Coppice Systems

Wastewater-fertigated short-rotation coppice systems (wfSRCs) have their origin in agroforestry tree plantations for producing biomass for either energy or material production. Many terms in the literature are used to describe plantations for producing woody biomass—namely, short-rotation coppice, short-rotation forestry, short-rotation willow coppice, short-rotation intensive culture, or short-rotation plantations. According to Kerr (2011), the main differences between short-rotation forestry and short-rotation coppice are the rotation period and the targeted end production [1]. The goal of short-rotation forestry with a rotation period of 8–20 years is to harvest timber or stem wood.

In SRC, fast-growing tree species are managed in short coppicing cycles (2–6 years and regenerated from the stools, which are expected to survive at least five rotation periods). Available research data relate to a variety of tested tree species (50 species) for SRC; however, the most important species used are *Salix* spp., *Populus* spp., *Eucalyptus* spp., *Acacia* spp., and *Gmelina arborea*.

SRC can be found in Europe (e.g., in all the Scandinavian countries, the Baltic countries, Germany, Poland, Ireland, England, and Italy) but also in North America and, on a smaller scale, in Asia, Latin America, and Australia. The estimated area of agroforestry tree plantations in Europe alone covers between 17 and 21 million hectares [2,3] and about 3 million hectares in the US [4].

Planting density in SRC varies, depending on tree species and soil quality. For willow planting, approx. 12–18 thousand willows per hectare are recommended, planted between 0.5 and 0.75 m apart, respectively, in a row. For poplar planting, 8–12 thousand plants per hectare are advised [5]. Operational yields of SRC in the Northern Hemisphere are 10–15 t DM/ha/y [6]. The woody biomass produced is mainly used as a renewable fuel source for heat and power generation, or for further processing into liquid biofuels.

The growth of these non-food/non-fodder crops depends on sufficient water and nutrient availability, which are often limited during the summer periods. Since wastewater comprises water and valuable nutrients, often in the right proportion for plant growth, nutrient and water demand can alternatively be met by applying pretreated wastewater, enabling sustainable nutrient and water recycling.

*1.2. Wastewater Treatment in SRC Systems (Vegetation Filter)*

Currently, investment and operating costs of existing technical wastewater treatment plants are high due to costs factors such as energy, applied chemicals, infrastructure (sewer system, machinery, and equipment), land, trained staff, and monitoring. The financial resources required and the need for trained staff are considerable implementation barriers for wastewater treatment systems in developing countries, especially in rural areas. At the same time, wastewater provides a resource of nitrogen (N), phosphorus (P), and water, which are the most limiting factors for plant growth. For example, N is a limiting parameter for willows growth as well as for other species, and therefore, these species can be used to bioremediate effluents with high N loads. A critical factor to maximise both the yield of wood and the efficiency of wastewater treatment is the N, P, and potassium (K) balance in the influent wastewater. According to Ericsson [7], the optimal ratio of nutrients for willow growth is N:P:K = 100:14:72, which is similar to the proportions of these nutrients in typical municipal wastewater [8].

Several European countries (e.g., Sweden, Poland, Denmark, and Estonia) have extensive experience in the application of wastewater in wfSRC systems and have demonstrated the high purification efficiency of willow wfSRCs [8–24]. In Denmark alone, more than 5000 zero-discharge willow wfSRC systems are currently operating. Despite a relatively large amount of information regarding the use of willows as vegetation filters, data mostly originate from a few northern European countries (mainly Sweden, Estonia, and Denmark). Information is scarce from other temperate regions where willows are successfully grown [25].

Additionally, other actors in different countries and climate zones (Australia, New Zealand, Ireland, Egypt, India, Greece, Canada, and the USA) [26–35] have implemented and tested wfSRC systems with different tree species as a multifunctional system for water treatment and biomass production.

In wfSRC systems, raw or pretreated wastewater is applied directly on the surface or in the upper soil layer with different irrigation systems. The treatment process occurs in the upper soil layer and in the root zone. Microorganisms present in the soil and the biofilms on the roots degrade the organic matter while the nutrients are mostly taken up by the plants or accumulated and transformed in the soil (Figure 1). WfSRC systems as other nature-based treatment solutions have relatively large footprints. Up to 15 m$^2$ of wfSRCs

are required to treat wastewater from one person, with a daily discharge of 100 L/d [22]. The application of wastewater to wfSRCs in the Northern Hemisphere is, in most cases, limited to the growing period of the trees, as the treatment efficiency and nutrient uptake by plants are reduced during the cold periods. In climates with long growing periods (10–11 months) and the absence of long periods of frost, wfSRCs can be run all year, as the soil–root filter system works and buffers nutrients.

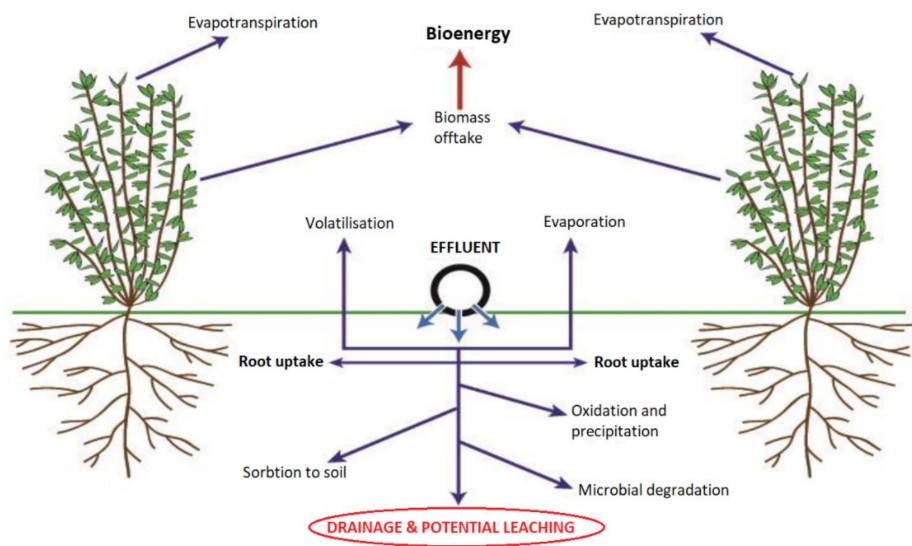

**Figure 1.** Processes involved in using woody plants for the phytoremediation of wastewater streams [36].

## 2. Materials and Methods

This study is based on the results provided by databases such as Scopus and Web of Science, using the following keywords "wastewater short-rotation coppice (plantation)", "nature-based wastewater treatment", "vegetation filter", "wastewater-based biomass production systems", and "phytoremediation" and limited to publications between 1972 and 2022. The analysed bibliometric data present an overview of the scientific activity regarding wfSRC systems worldwide. The found data were used to build a database to produce reliable information on the state of the art including scientific activities and results of implementation trials in specific regions. The bibliometric report cannot give an absolute overview of each region or country due to limiting factors such as language and accessibility of data.

Experiences from 135 papers were reviewed, and each paper was labelled according to its key topic. Both full-scale and pilot systems were included in the study. In order to identify suitable pilots, tests, and designs, the grey literature was also reviewed. This included project material of three major European research projects on this topic—namely, BIO-PROS [5], WACOSYS [37], and PAVITR [38]. The details of the literature analysis are reported in the Table S1.

In total, 124 publications were considered for the review of the state of the art, as some papers dealt with economic and social issues. From these 124 publications, 47 provided relevant numerical data on water treatment efficiency and biomass production, with wfSRC systems in 31 countries (Figure 2). Different designs, tree species, planting schemes, geographic locations, and climate zones for the implementation of wfSRCs were identified. Only larger experimental-scale, pilot-scale, and full-scale outdoor trials receiving real wastewater were considered in this review. To present the performance of systems, the gathered and processed information from the 47 papers includes (1) location and climate, (2) design and capacity, (3) tree species used, (4) type of wastewater, (5) organic loading rate (BOD, COD), (6) total nitrogen (TN) concentrations in the influent and effluent, (7) total phosphorus (TP) concentrations in the influent and effluent, and (8) biomass production.

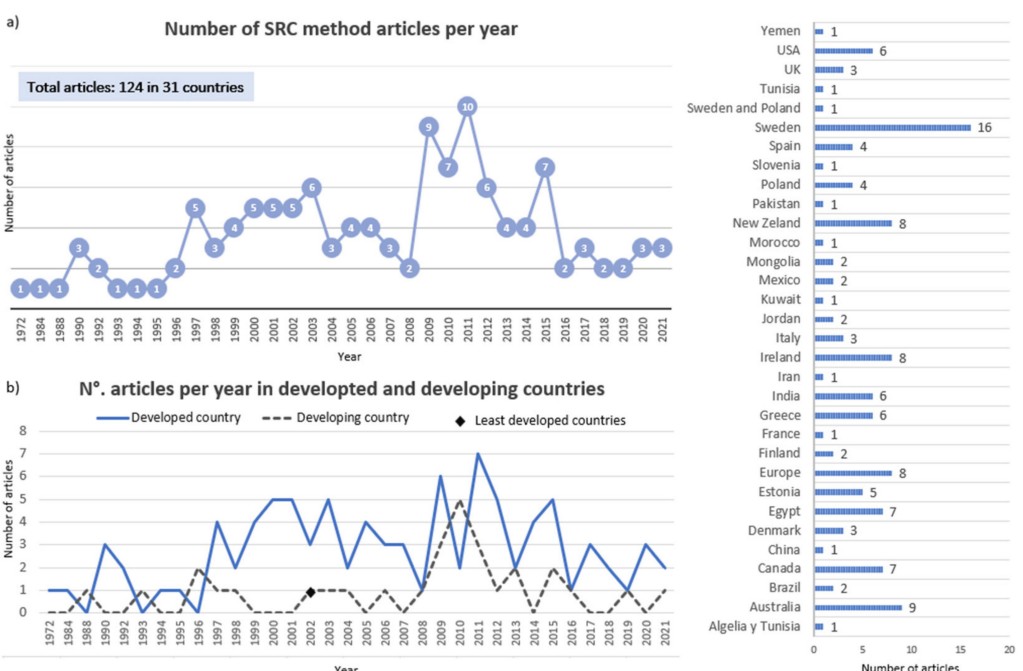

**Figure 2.** (**a**) Global numbers of scientific publications related to wfSRCs from 1974 to 2021; (**b**) wfSRC publications in developed and developing countries between 1974 and 2021.

Figure 2a shows the number of scientific papers available in the Scopus database and other important database platforms related to scientific articles corresponding to the keywords listed in the first paragraph of this section. The research activity shows a varying number of articles over the last few decades. In fact, there is a large increase from 2009 to 2012, underlying a growing interest in this topic. Figure 2b shows the comparison between numbers of publications in developed economies and developing economies from 1974 to 2021. According to our findings, 86 articles (73%) originate from developed economies and only 32 articles (27%) from developing economies.

## 3. Results and Discussion

### 3.1. Types of Wastewater-Fertigated Short-Rotation Coppice Systems

Different kinds of wfSRC systems were found in the literature, including the following systems:

Impermeable zero-discharge systems (Figure 3): In these wfSRC systems, the basins are excavated to a depth of approx. 1.8 m. A geosynthetic barrier and an impermeable membrane are laid at the bottom and the sides of the excavated basin, to prevent infiltration to groundwater and leakage of water from the bed to the surroundings. The excavated soil is then back-filled into the basin, up to ground level, in addition to installing a distribution layer [29]. These systems can be operated as zero-discharge systems, in which all wastewater and rainwater entering the system have to be evapotranspired to the atmosphere. An alternative operation mode allows infiltration or discharge via a drainage system (e.g., willow evapotranspiration systems in Denmark). Details are well-described in [10,29,39,40].

Impermeable wfSRC systems with discharge: these types of wfSRC systems collect all excess water such as lysimeters filled with sand and clay and planted with trees.

Permeable wfSRC systems with water infiltration (Figure 4): In permeable wfSRC systems, trees are planted and fertigated on land which is not sealed with an impermeable membrane, allowing water infiltration. Applied wastewater is treated in the upper soil layer and root zone. Excess water is infiltrated into groundwater.

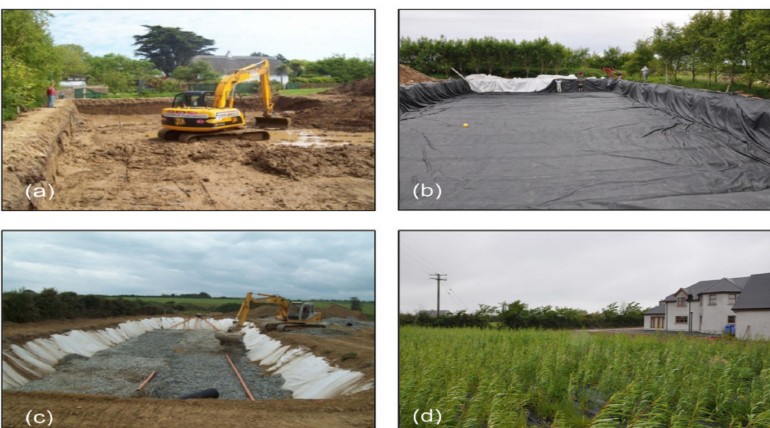

**Figure 3.** Stages in the construction of an evapotranspiration system: (**a**) excavation of basin (**b**) laying the impermeable membrane (**c**) installation of distribution layer, and (**d**) initial growth of the willows [29].

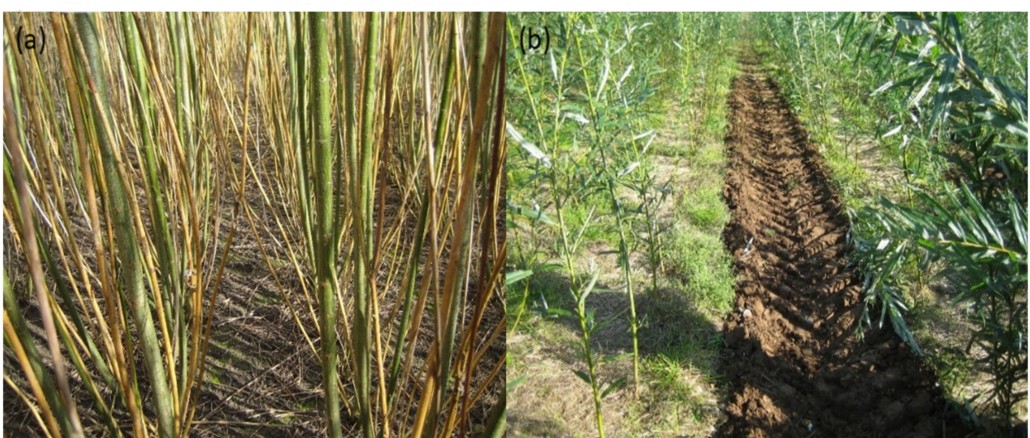

**Figure 4.** (**a**) Open wfSRC system (poplar) in Granada (Spain), (photo: Haenel 2008); (**b**) open wfSRC system (willow) in Enköping (Sweden), (photo: Haenel 2008).

### 3.2. Size of wfSRC Systems

The size of wfSRC systems varies according to the scale and the objectives of trials. The surface areas range from small systems of 1 m$^2$ (lysimeters planted with poplar and willows) to small setups with up to 30 m$^2$, to medium systems of 150–200 m$^2$, to large-scale implementations of up to hundreds of hectares. The largest application has been found in Huolinguole City, China (primary treated wastewater for summer irrigation of different tree species including *Larix* sp., *Pinus* sp., and *Populus* sp. at 880 ha [41].

### 3.3. Tree Selection

There are specific requirements for the selection of suitable tree species for wfSRCs such as high water demand and evapotranspiration rates, high biomass yield and coppicing ability, tolerance to high nutrient concentration and high water saturation in the soil, high filtering capacity and nutrient uptake, shallow root system and good ability to promote denitrification in the root zone, and selective uptake of heavy metals such as cadmium [17,42]. The main tree species found in wfSCR are willows (*Salix* spp.) in the northern part of Europe (Scandinavia, the British Islands, Ireland, the Baltic States) and the US. The main reasons given are that willow is a fast-growing and water-demanding species, can use relatively high levels of N and P, and is highly tolerant to various contaminants including some heavy metals [43].

In many parts of the US, Canada, and southern Europe (e.g., Italy, Spain, and Greece), diverse poplar varieties are widely used for wfSRC. In Australia, New Zealand, and South Asia, there is a greater predominance of eucalyptus and other tree species such as *Pinus radiata* and *Casuarina* spp. In western Asia and Africa, the use of other tree types predominates (*Olea* spp., *Citrus* spp., *Tamarix* spp., *Casuarina* spp., *Acacia* spp., among others), followed by *Eucalyptus* spp.

Besides choosing appropriate tree species, clone selection also plays an important role in the system's performance. Different clones of listed species have been selected for higher biomass production through targeted breeding programmes in different countries.

### 3.4. Planting Density

The optimum planting density to obtain maximum yield and treatment performance varies, depending on tree species, soil type, climate, and wastewater characteristics. For the most common tree species (willow), recommendations start from 15,000 plants/ha in Estonia [44] up to 25,000 plants/ha in the UK [45–47]. For poplar plantations, a density of 15,000 cuttings per ha should not be exceeded. Higher densities of around 20,000–25,000 cuttings/ha are more productive in the first year, but the effect of competition reduced the differences, as rotation approached three years [48].

Besides optimum planting density for a given environment and tree species, the planting scheme can vary also according to socio-economic factors such as available budget, time schedule, and main objectives of the wfSRC system (treatment efficiency versus biomass production).

### 3.5. Overview of Used Tree Species and Geographical Location

Based on the analysed data, *Salix* spp. has been the most studied genus in wfSRC systems (32%), followed by *Eucalyptus* spp. (21%), *Populus* spp. (18%), *Pinus* spp. (6%), *Acacia* spp. (5%), *Casuarina* spp. (3%), *Citrus* spp., and *Olea* spp., each with 2%. There are also data available for plantations fertigated with wastewater for *Dalbergia* spp., *Musa* spp., *Sambucus* spp., *Betula* spp., *Larix* spp., *Acer* spp., *Fraxinus* spp., *Platanus* spp., *Millettia* spp., *Melia* spp., *Alstonia* spp., *Tamarix* spp., *Swietenia* spp., *Conocarpus* spp., *Khaya* spp., and species from the bamboo family. This confirms general statements found in the literature that, in northern European climates, the most widely used tree species are willows (*Salix* spp.), whereas poplars (*Populus* spp.) and/or eucalyptus (*Eucalyptus* spp.) are mostly used in southern climates [49] (Figure 5).

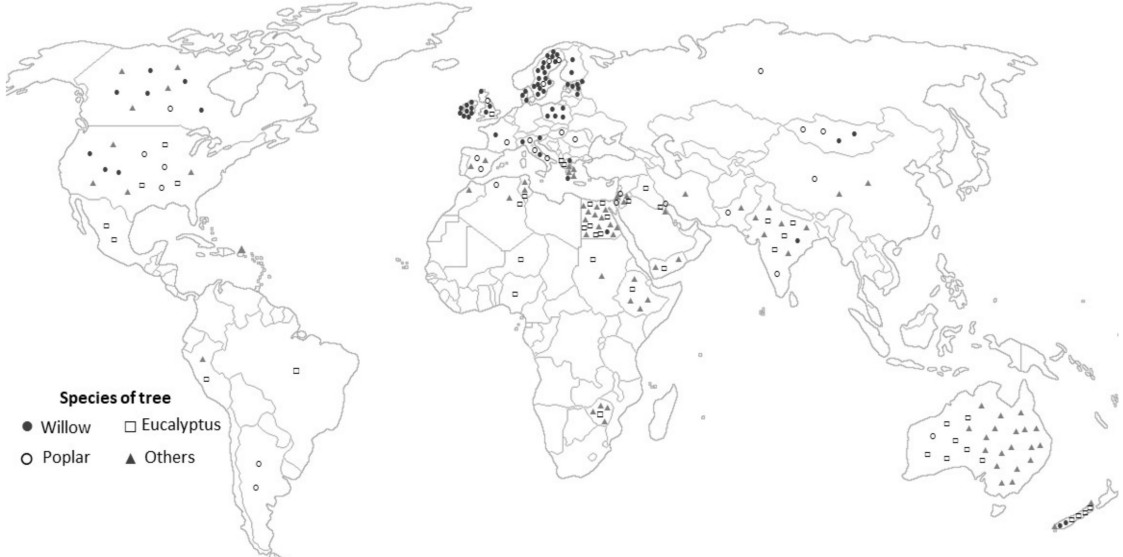

**Figure 5.** Locations by country of reviewed wastewater-fertigated short-rotation coppice systems.

*3.6. Geographical Distribution*

A good number of data and experiences are available for wfSRCs in many developed countries, especially in the Northern Hemisphere (Europe, US) but limited amounts of accessible information from developing countries. In contrast, the highest application potential for wfSRCs seems to be in developing countries due to suitable climates and missing wastewater treatment infrastructure, especially in rural areas.

*3.7. Treatment Performance*

The treatment performances of reviewed wfSRC systems depend on factors such as climate; soil type (e.g., buffer and percolation capacity); properties of the selected tree species (e.g., water demand, evapotranspiration rate, nutrient-use efficiency); organic, nutrient and hydraulic loads of applied wastewater; and the design and operation of the system. Variable inlet conditions (e.g., wastewater application only during the growth period) may also affect removal efficiency and system behaviour. In general, wfSRC systems are characterised by a high removal potential for pathogens due to filtration, adsorption onto soil particles, and exposure to various abiotic stresses [50–53]. In the studied cases, the applied wastewater originated from domestic sources (85%), followed by various industries—namely, bakery, meat and dairy production (8%), and landfill leachate (7%).

Additionally, the wastewater differed regarding the treatment steps prior to application to wfSRCs (primary-, secondary-, tertiary-, or fully treated) and, thus, had variable concentrations of nutrients, organics, and other pollutants. In this paper, treatment performance was evaluated in terms of conventional pollutants, i.e., chemical oxygen demand (COD), biochemical oxygen demand ($BOD_5$), total nitrogen (TN), and total phosphorous (TP). Due to different designs, local regulations, targeted pollutants, and applied wastewater, a direct comparison in terms of treatment performances was not performed.

Organic matter of the applied wastewater is removed in wfSRCs through microbial decomposition, deposition, and filtration; nitrogen is removed by ammonification, nitrification, denitrification, plant uptake, and matrix adsorption; phosphorus is removed by adsorption, plant uptake, complexation, and precipitation [24]. The root system is reported to take up 75–95% of the nitrogen and phosphorus in the wastewater [20].

The removal of harvested biomass is an important pathway for nutrient recovery since harvested biomass can be used as a soil amendment as well as a slow-release fertiliser. Recovery of up to 650 and 100 kg/ha of N and P, respectively, has been reported for woody species [28,54,55].

3.7.1. Biochemical Oxygen Demand (BOD) and Chemical Oxygen Demand (COD)

In the reviewed cases, organic loads varied greatly, depending on type of wastewater and level of pretreatment. The organic loads applied to wfSRCs receiving domestic wastewater fluctuated from 104 mg/L or 7.3 kg $BOD_5$/ha/d [56] to 122 mg/L or 1146 kg $BOD_5$/ha/d in China [41]. In contrast, for industrial wastewater (dairy farm effluents), according to Forbes et al. (2017), the influent concentrations varied from 230 mg/L or 16.1 kg $BOD_5$/ha/d to 6600 mg/L or 463.3 kg $BOD_5$/ha/d. According to Paranychianakis (2016) [53], the organic load should not, in the long run, exceed 500 kg $BOD_5$/ha/d for wfSRC. In general, the reviewed results highlighted that wfSRCs can successfully grow under high organic loads.

COD and BOD removal data were available from 18 reviewed systems, with 37 datasets in total, of which 22 datasets were related to BOD (Figure 6) and 15 to COD (Figure 7). COD and BOD removal efficiency varied from 46% to 96% and from 60% to 99%, respectively. Overall, 81% of the datasets (30 datasets) showed removal efficiency higher than 80% for both parameters. The highest BOD removal efficiency (99%) was reported in Ireland on a willow wfSRC receiving cattle farm wastewater [46], followed by a system in Quebec, Canada, on a willow wfSRC treating municipal wastewater, with 98% BOD removal [57,58].

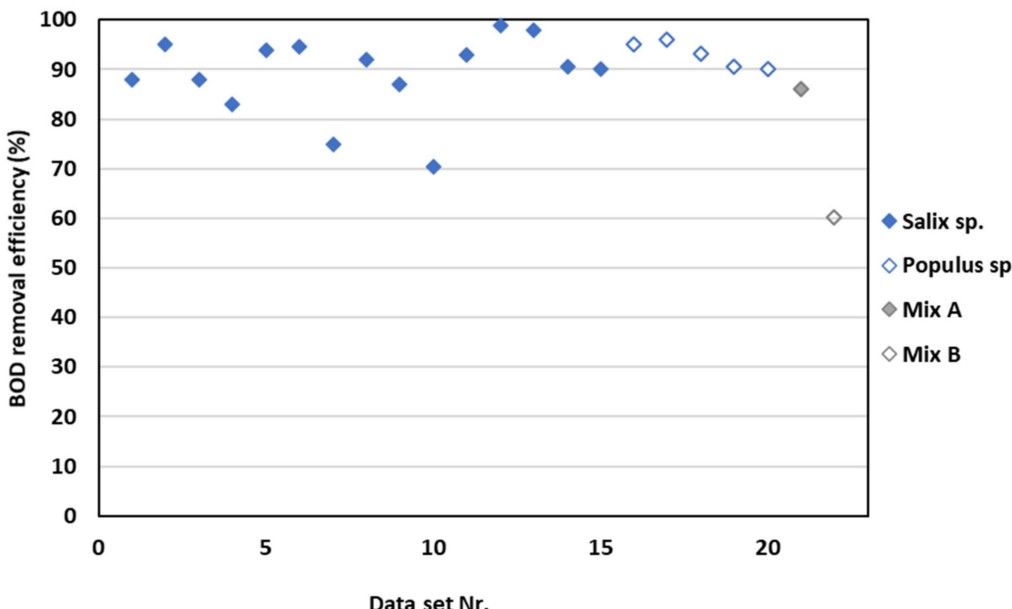

**Figure 6.** BOD removal efficiency of wfSRCs according to species. Mix A includes *Larix principis rupprechetii, Pinus silvestris, Populus xoeichenesis*, and mix B includes *Populus deltoides, Eucalyptus* sp., *Salix alba, Melia azedarach*. Data sources from [8,24,41,46,56,57,59–66].

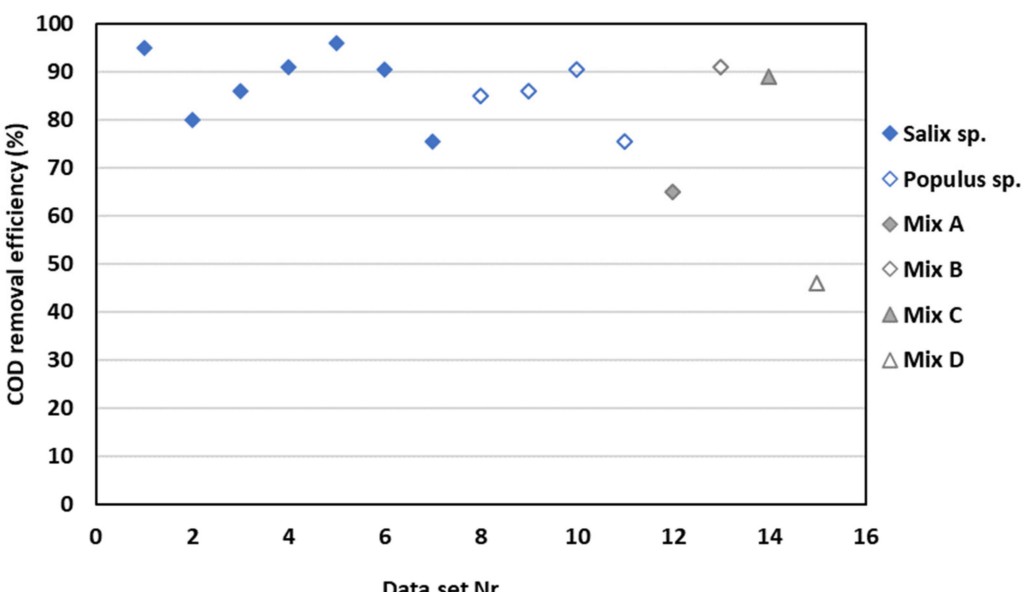

**Figure 7.** COD removal efficiency of wfSRCs per species. Mix A includes *Larix principis rupprechetii, Pinus silvestris, Populus xoeichenesis*; mix B includes *Eucalyptus camandulensis, Acacia cyanophylla, Populus nigra*; mix C includes *Eucalyptus camandulensis, Acacia cyanophylla, Populus nigra, Arundo donax*; and mix D includes *Populus deltoides, Eucalyptus* sp., *Salix alba, Melia azedarach*. Data sources from [41,51,56–58,62,63,65–68].

The lowest COD and BOD removal efficiency was reported in Pantnagar, India, on an experimental wfSRC system planted with *Eucalyptus* sp., *Poplar* sp., *Salix* sp., and *Melia* sp. treating greywater with removal efficiencies of COD and BOD of 60% and 46%, respectively. The reported removal efficiency was greatly influenced by plant growth, inflow concentration, and variation in climatic conditions [66].

### 3.7.2. Total Nitrogen (TN)

Nitrogen discharge to the environment, including nitrate leaching to groundwater and ammonia discharges, are of major concern in wfSRC systems.

From the review, it was revealed that 43 wfSRC papers reported on TN removal. Most of the wfSRC trials did not measure TN due to a lack of demanded legal limits for TN discharge in many countries and regions. There are large differences in the documented TN removal efficiency, ranging from 12% to 99% depending on the applied TN load. In total, 6 trials reported a TN removal higher than 90%, 22 trials reached a TN removal efficiency of 50%–90%, and in 15 trials, the removal was below 50% (Figure 8). The three highest TN removal efficiencies were documented in Ireland (willow wfSRC treating dairy farm effluent), with 99% [69]. A trial in Canada (poplar wfSRC, treating municipal wastewater) reached a removal of 96% [70], and a trial in Sweden (willow wfSRCs with municipal wastewater and sludge) documented a removal efficiency of 96% [15].

Willows were apparently very effective in removing N from wastewater even at high N loading rates; specifically, willow wfSRCs can treat TN loading of more than 200 kg/ha/y, due to denitrification and long-term build-up of nitrogen in the soil [59]. A comparison of experimental data from full-scale willow wfSRC fields in central Sweden suggests that the N load that can be treated is considerably higher than the N requirements of willow SRC and depends on site-specific conditions [71].

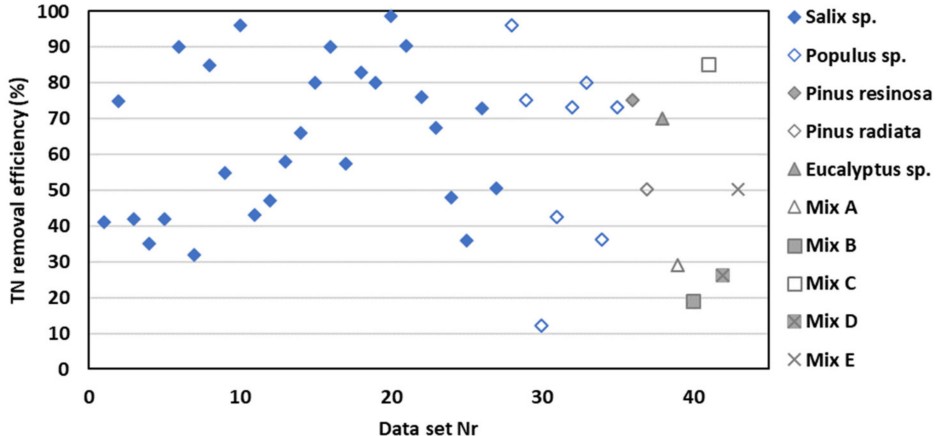

**Figure 8.** TN removal efficiency of wfSRCs per species. Mix A includes *Eucalyptus* sp., *Populus* sp., *Arundo donax*, *Salix alba*, *Melia azedarach*, and *Acacia cyanophylla*; mix B includes *Eucalyptus camadulensis*, *E. grandis*, *E. saligna*, *Casuarina cunninghamiana*, *Pinus radiata* D. Don, *Populus deltoides*; mix C includes *L. principis-rupprechetii*, *Pinus sylvestris* and *Populus xoechenesis*; mix D includes *Eucalyptus* sp., *Acacia* sp., *Populus* sp.; and mix E includes *Populus deltoides*, *Eucalyptus*, *Salix alba*, *Melia azedarach*. Data sources from [8,15,20,22,24,25,27,29,35,41,42,46,56–61,63–67,72–81].

On the other hand, even higher TN loading rates result in reduced treatment efficiency, e.g., Kowalik and Randerson, (1993) [7] report treatment efficiency of less than 50% at TN loads of 1120 and 2240 kg N/ha/y. In addition, another critical factor for TN removal is the accumulation of nutrients in the biomass. This varied significantly among species. Therefore, a selection of appropriate species with high N demand can increase TN removal. Most importantly, in many cases, nitrification is the limiting factor for nitrogen removal because of the deficiency of oxygen that hinders the process. Better aeration (air circulation), selection of suitable tree species, correct selection and equal dosage of wastewater, longer retention time, and extended area or aerobic pretreatment can increase nitrification and overall nitrogen removal efficiency in wfSRC systems.

### 3.7.3. Total Phosphorus (TP)

In 31 of the reviewed papers, the reported results concerned influent and effluent TP concentrations, for which 41 datasets were obtained. TP removal efficiency varied

from 14% to 100%. Half of the documented trials showed a TP removal rate of more than 80%. Additionally, one-third of the trials documented a TP removal range between 40% and 80%, and only 15% recorded less than 40% (Figure 9). The highest TP removal efficiency was reported in Ireland on willow planted cylindrical containers irrigated with secondary treated effluents with 100% removal of TP [73]. In secondary-treated wastewater, the organic N and P are already mineralised; therefore, the availability of nutrients for plant uptake is higher than in primary-treated wastewater, which can result in higher treatment efficiency.

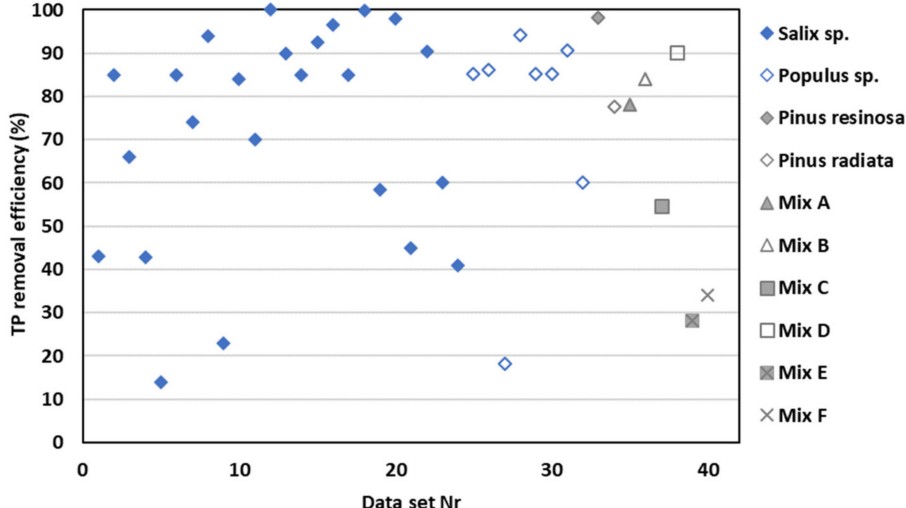

**Figure 9.** TP removal efficiency of wfSRCs per species. Mix A includes *Eucalyptus* sp., *Populus* sp., *Arundo donax*, *Salix alba*, *Melia azedarach*, and *Acacia cyanophylla*; mix B includes *L. principis-rupprechetii*, *Pinus sylvestris*, and *Populus xoechenesis*; mix C includes *Eucalyptus grandis* and *Pinus radiata*; mix D includes *Eucalyptus camandulensis*, *Acacia cyanophylous*, and *Populus nigra*, *A. donax*; mix E includes *Eucalyptus* sp., *Acacia* sp. and *Populus* sp.; mix F includes *Populus deltoides*, *Eucalyptus* sp., *Salix alba*, and *Melia azedarach*. Data sources from [8,15,20,22,24,25,29,35,41,42,46,56,58–60,63–68,72–76,78–80,82].

In Aarike, Estonia, low average efficiencies of 14% for TP and 8% for $PO_4$-P were documented. The system used was a surface flow system with very short retention and contact time. The wastewater originated from a dwelling house, with average concentrations of total-P and $PO_4$-P in the inflow varying from 3 to 15 mg/L and from 2.5 to 12 mg/L, respectively. Typically, most of the phosphorus is adsorbed by soil particles, thus needing a longer retention time [24].

From the gathered results, it is clear that well-designed and -managed wfSRC systems are suitable for reliable TP removal. Important factors for a high removal rate seem to be the size of the wfSRC area in relation to the loading, loading rate, selected tree species, soil properties, and resulting retention time in relation to the incoming TP load.

### 3.7.4. Total Potassium (K+)

As Potassium is not a key pollutant, and its concentrations in effluents from domestic wastewater sources are relatively low (up to 20 mgL$^{-1}$) [83], only a few of the reviewed trials measured total potassium. Data reported in Spain for poplar showed a high removal efficiency for K+ (90%) [67], but in colder climates (Finland), removal efficiency as low as 25% was reported [74]. However, potassium has a low leachability and, therefore, high potential to be accumulated in the soil. In Estonia, increased soil K+ concentration was documented in two wastewater irrigation seasons (from 97 to 135 mg kg$^{-1}$ dw) [72]. In Ireland, soil K+ content also increased significantly in a 4-year trial [46].

### 3.7.5. Pathogens, Micropollutants, and Heavy Metal Removal

Factors such as oxidation, sunlight exposure, filtration, desiccation, and antagonism with microbial fauna in the upper soil layer lead to a massive reduction of pathogens [84]. In trials using *E. coli* as an indicator, substantial pathogen removal in wfSRCs was recorded [60]. In Greece, no faecal coliforms were detected in 30 cm soil depth after one year of wastewater application [78].

In the reviewed studies, the capacity of wfSRC systems to degrade micropollutants in the wastewater such as antibiotics and hormones was considered only in exceptional cases. In a field trial in Spain with poplars, the removal efficiency for all selected emerging organic contaminants was higher than 90% with more than 90% of the selected emerging organic contaminants removed [85].

Only in a few of the documented trials were the removal efficiencies of heavy metals in wfSRCs reported, as typical concentrations of heavy metals in municipal wastewater are expectedly low. In general, studies on the distribution of heavy metals in wfSRCs did not report elevated heavy metal concentrations in the topsoil layer. The main reason given is leaching into surrounding water bodies or into the subsurface [86]. However, zero-discharge systems with high loads of heavy metals showed a slight increase in soil concentrations after several growing seasons [75,87]. The harvested biomass can also contain heavy metals such as Cd and Zn, as some willow clones are able to absorb them efficiently [88,89].

### 3.8. Biomass Production

The results on biomass production of the present review showed significant differences in woody biomass yield per area. Yields reported in wfSRC systems are influenced by tree species, clone selection, planting density, harvesting cycle, climate and soil conditions, and water and nutrient content of the supplied wastewater. These factors make the comparison of data among different trials very challenging.

In general, enhanced biomass growth in wfSRC systems due to wastewater fertigation was reported in all reviewed documents. Biomass increases in wfSRCs are linked to the additional supply of nutrients and irrigation water. Thus, the application of nutrient-rich wastewater offers an alternative to reduce both mineral fertiliser and freshwater consumption, as reported in, e.g., [15,90].

In the reviewed cases, the reported harvested woody biomass varied from 3.7 to 40 t DM/ha/yr (Figure 10). In willow wfSRC, wastewater fertilisation increased biomass production by 4–8 t DM/ha/yr, meaning an increase in yields between 20% and 100%, compared with average yields for well-managed, non-irrigated willow plantations on fertile soils with good permeability [20,49]. In eucalyptus wfSRC, an increase of 83% in biomass productivity, compared with traditional cultivation, was reported [91].

The highest biomass production in wfSRC systems was reported in Australia, in a eucalyptus wfSRC-applying domestic wastewater, reaching 40 t DM/ha/y [92]. The lowest biomass production in a wfSRC system was reported in northern Finland—namely, 3.7 t DM/ha/yr, in a pilot willow wetland. However, it was also concluded in this case that willow biomass production was higher when fertigated with wastewater than in a reference area with no sewage irrigation [74].

Of the total 56 reviewed datasets, eucalyptus showed the highest biomass production results, followed by poplar.

In general, the application of wastewater in wfSRCs was found to increase biomass production as well as decrease production costs. An annual municipal wastewater load of 600 mm/year, containing N:P:K = 100:20:65 supplied the required water, in addition to fulfilling the requirements of N, P, K, and other macronutrients [9]. By applying nutrient-rich wastewater, conventional fertilisation practices can be reduced or even discontinued. It was estimated that 7–20 Euros could be saved per kilo of N by using natural instead of mineral fertilisers. The production costs of woody biomass can, therefore, be reduced by 20–30% [16].

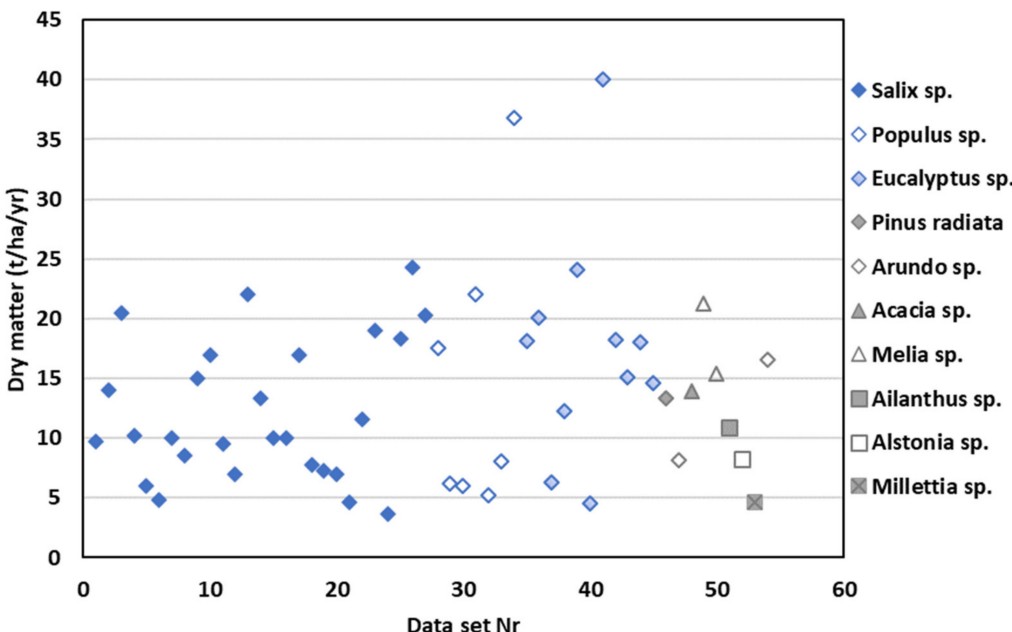

**Figure 10.** Biomass production of wfSRC per species. Data sources from [8,20,24,25,28,29,31,39,42,43, 46,57,59–61,63,64,66,68,73–77,79,82,90–92].

The main documented use of biomass produced in wfSRCs is for energy purposes. Depending on end-user requirements, availability of machinery, and drying facilities, wfSRC biomass is harvested as wood chips, rods, or billets [5]. Alternative known uses are for the construction industry or as packaging material [37]. Other potential applications are raw materials for the chemical industry and fibre-based building materials.

## 4. Conclusions

Based on the findings of this review, wfSRCs have a high potential to be viable and cost-effective wastewater treatment alternatives to conventional treatment plants. Scientific studies indicate the high removal potential for BOD, COD, TN, TP, and enhanced biomass production. The approach might have also the potential to treat micropollutants, but this needs to be confirmed in upcoming research trials.

The majority of the 124 reviewed wfSRC case studies demonstrated superior performance in nutrient and organic matter removal, presenting microbial degradation, oxidation, and plant uptake as the main removal mechanisms. Zero-discharge systems, in particular, accomplished a 100% nutrient and organic matter removal efficiency, as the incoming water is fully evapotranspired, avoiding any discharge from the system. In addition, the evaluated systems proved to be highly efficient in the removal of microorganisms.

It can be concluded that one hectare of a well-designed and -operated wfSRC system can treat a load of up to 200 kg BOD/ha/day in the long term, handling 3300 population equivalents. The resulting woody biomass production of more than 10 t DM/ha/yr (up to 40 t/ha/yr were documented) are exit gates for the incorporated nutrients and additional income sources for the system operators. Based on the trials documented in the reviewed literature, favourable factors for a successful application of wfSRCs are suitable climate conditions (long vegetation periods, short and mild winters), availability of appropriate wastewater sources, sufficient land area with suitable soils (medium soil buffer capacity), and a well-balanced selection of local tree species for a high treatment performance and production of marketable biomass. Additionally, professional planning, design, setup, and operation of wfSRC systems are essential in order to guarantee a high treatment performance, as well as high biomass production.

This review highlighted significant differences between plant species in terms of treatment efficiency, biomass production, and nutrient removal. At present, diverse willow

species in colder climates and poplar species in warmer climates have shown convincing results. On the other hand, trials with species such as eucalyptus and bamboo indicate tremendous potential as well. Thus, by an improved selection process and suitable measures such as specific breeding programmes of suitable plant species for wfSRC, the treatment capacity, but also the cost–benefit ratio, could be substantially improved.

To avoid environmental risks such as groundwater contamination with nitrates, especially in ecologically sensitive areas with shallow aquifers, additional management schemes should be adopted to close the gap between nutrient load and the potential of wfSRCs for nutrient removal. Possible actions could include adjustment of wastewater application rates to the wfSRC system capacity, the vegetation cycle, use of high-accumulating tree species, adjustment of the rotation period, and the implementation of suitable pretreatment schemes to increase the availability of nutrients.

This review also showed a great imbalance between suitable locations and conditions for the application of wfSRC systems which can be found in many developing countries in Asia, Latin America, and Africa, and locations of actual implemented and ongoing R&D activities in this field (mainly Europe and North America). Most reliable data and resulting guidelines and recommendations on suitable plant species, treatment procedures, biomass production potentials, and cost–benefit calculations are available only for regions and climates in Europe and North America. Considering the lack of existing wastewater treatment infrastructure, favourable climatic conditions, and large and unexploited biodiversity of suitable tree species, there is a unique application potential for wfSRC, especially for rural areas in many developing countries. In these mostly sparsely populated rural regions, wfSRCs are low-cost and efficient alternatives to the construction of cost-intensive, high-standard technical treatment of wastewater. In well-managed wfSRC systems, the purification rates are higher than those in many conventional treatment facilities, reducing the release of nutrients to water bodies.

By producing and marketing valuable woody biomass, the treatment process of selected wastewater types could be turned from a cost-intensive to a profitable, highly productive scheme. Thus, the large-scale demonstration of wfSRC "light-house" projects in well-suited locations in developing countries will gain interest among local stakeholders such as farmers and remove implementation barriers and could be a "game changer". To reach a broadly accessible market, existing gaps in research such as those regarding investigations on long-time impacts on soil and groundwater, management of micropollutants, breeding programmes for suitable tree species and clones, high-value uses of biomass (biorefinery products, fibres, etc.), nutrient recovery and interaction with food and fodder production, need to be addressed by the international scientific community.

**Supplementary Materials:** The following are available online at http://www.mdpi.com/1999-4907/13/5/810/s1, Table S1: A detailed literature analysis.

**Author Contributions:** Conceptualisation, M.H., H.B. and C.A.A.; methodology, M.H. and C.A.A.; validation, M.H., D.I., H.B. and C.A.A.; formal analysis, M.H., D.I. and C.A.A.; investigation, M.H.; resources, M.H.; data curation, M.H., D.I. and C.A.A.; writing—original draft preparation, M.H.; writing—review and editing, M.H., D.I. and C.A.A.; visualisation, M.H., D.I. and C.A.A.; supervision, C.A.A. and H.B.; project administration, M.H.; funding acquisition, M.H. All authors have read and agreed to the published version of the manuscript.

**Funding:** This study was co-financed by the EU-INDIA project "PAVITR": 821410–H2020.

**Data Availability Statement:** The data presented in this study are available on request from the corresponding author.

**Conflicts of Interest:** The authors declare no conflict of interest. The funders had no role in the design of the review; in the collection, analyses, or interpretation of data; in the writing of the manuscript; or in the decision to publish the findings.

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
