# Peer review of "Wastewater-Fertigated Short-Rotation Coppice, a Combined Scheme of Wastewater Treatment and Biomass Production: A State-of-the-Art Review"

_forests, doi:10.3390/f13050810_

Round 1

Reviewer 1 Report

I don't have comments.

Author Response

Thanks for the review. 

Reviewer 2 Report

The paper is focus on the state-of-the-art of wastewater fertigated SRC systems (wfSRC) worldwide to facilitate the use of wfSRC systems around the world and especially in developing countries. This review has a relatively complete summary and analysis on Wastewater fertigated Short rotation coppice – a combined scheme of wastewater treatment and biomass production. However, there are still some minor problems that need further improvement.

  1. Line25-27, this review describes that the performance review includes the standard water quality parameters BOD5, COD, nitrogen, phosphorous, and potassium as well as pathogen and emergent contaminant removal and biomass production rates in the abstract. However, Potassium and pathogens were not mentioned in the analysis and discussion. Please add and improve.
  2. The abstract could be further refined.
  3. Whether there are specific references corresponding to data analysis about the standard water quality parameters BOD5, COD, nitrogen, phosphorous as well as emergent contaminant removal and biomass production rates in Fig6-Fig10?
  4. The references cited in this paper should pay attention to their influence and time.

Author Response

To 1. A new paragraph and related references on potassium (K+) have been added (3.7.3). Also, additional material regarding pathogen removal results have been integrated and the paragraph 3.7.4 has been renamed and extended (3.7.4 Pathogens, micropollutants and heavy metal removal).

To 2. The abstract has been refined and shortened (less then 200 words).

To 3. All references used to plot the graphs in figs 6-10 have been added under each graph ("data sources from [xx, yy-zz]".

To 4. Additional actual publications have been added to the references. Older references are of significant importance and the data are still relevant.

General: Author Contributions, Funding, Data Availability Statement and a declaration of Conflicts of Interest have been added.

Reviewer 3 Report

Dear Authors,

The subject of the study is interesting and topical, with high scientific and practical importance.

Some suggestions and corrections were made in the article.

The following aspects are brought to the attention of the authors.

1.

Keywords

According to Instructions for Authors, and Microsoft Word template, Forests journal, it is recommended ";" between keywords.

"Keywords: keyword 1; keyword 2; keyword 3"

Revision is recommended

2.

Abstract

Instructions for Authors, Forests journal, recommends “Abstract: A single paragraph of about 200 words maximum.”

According to this recommendation, the Abstract is a bit more extensive and a revision would be appropriate.

3.

Bibliographic sources citation

eg

Page 1, row 48-49

Bibliographic source “Hardcastle et al. (2006)” is quoted in the text, but is not found in the References chapter.

It is the first bibliographic source in the article, and should be number 1 in the References chapter, and in the article content [1].

It is recommended to check first.

Requires revision and correction, as appropriate

All other bibliographic sources will need to be renumbered.

Therefore "[1,2]" on page 2, row 60, will become "[2,3]".

Similarly, the other bibliographic sources should be renumbered, in the content of the article, according to the References chapter.

4.

Bibliographic source cited twice

Eg.

Page 8, row 271

"Forbes et al. (2017)"

References chapter

Page 16, rows 603 - 605

”44. E. G. A. Forbes, C. R. Johnston, J. E. Archer, and A. R. McCracken, “SRC willow as a bioremediation medium for 603 a dairy farm effluent with high pollution potential,” Biomass and Bioenergy, vol. 105, pp. 174–189, 2017, doi: 604 10.1016/j.biombioe.2017.06.019.”

Page 17, rows 633-635

”55. E. G. A. Forbes, C. R. Johnston, J. E. Archer, and A. R. McCracken, “SRC willow as a bioremediation medium for 633 a dairy farm effluent with high pollution potential,” Biomass and Bioenergy, vol. 105, pp. 174–189, Oct. 2017, doi: 634 10.1016/j.biombioe.2017.06.019.”

5.

Conclusions

The conclusions are too broad.

It is recommended to synthesize and revise the content in the Conclusions chapter, in accordance with Instructions for Authors, Forests journal.

6.

Author Contributions

Data Availability Statement

Acknowledgments

Conflicts of Interest

It is recommended to clarify and complete according to the Instructions for Authors, Forests Journal.

7.

References

The entire References chapter needs to be revised, according to the Instructions for Authors, and the Microsoft Word template, Forests journal.

“Author 1, A.B.; Author 2, C.D. Title of the article. Abbreviated Journal Name Year, Volume, page range.”

eg.

”Özdemir, E.D.;  Härdtlein, M.; Eltrop, L. Land substitution effects of biofuel side products and implications on the land area requirement for EU 2020 biofuel targets. Energy Policy 2009, 37(8), 2986–2996. doi: 10.1016/j.enpol.2009.03.051.”

Instead of

”E. D. Özdemir, M. Härdtlein, and L. Eltrop, “Land substitution effects of biofuel side products and implications on the land area requirement for EU 2020 biofuel targets,” Energy Policy, vol. 37, no. 8, pp. 2986–2996, 2009, doi: 10.1016/j.enpol.2009.03.051.”

Some bibliographic sources cited in the text were not found in the References chapter.

Eg

Page 1, row 48-49

Bibliographic source “Hardcastle et al. (2006)” is quoted in the text, but is not found in the References chapter.

Other bibliographic sources cited in the text have been cited twice in the references chapter (numbered twice) which has influenced the number of bibliographic sources.

Bibliographic source cited twice

eg

Page 8, row 271

Forbes et al. (2017)

References

Page 16, rows 603 - 605

”44. E. G. A. Forbes, C. R. Johnston, J. E. Archer, and A. R. McCracken, “SRC willow as a bioremediation medium for 603 a dairy farm effluent with high pollution potential,” Biomass and Bioenergy, vol. 105, pp. 174–189, 2017, doi: 604 10.1016/j.biombioe.2017.06.019.”

Page 17, rows 633-635

”55. E. G. A. Forbes, C. R. Johnston, J. E. Archer, and A. R. McCracken, “SRC willow as a bioremediation medium for 633 a dairy farm effluent with high pollution potential,” Biomass and Bioenergy, vol. 105, pp. 174–189, Oct. 2017, doi: 634 10.1016/j.biombioe.2017.06.019.”

It is recommended to check each bibliographic source to be the correspondence between the sources cited in the text and those in the References chapter.

Author Response

Dear reviewer, 

thanks for your comments. The following tasks have been done.

1.Keywords have been revised. 

To 2. Abstract has been revised (less than 200 words)

To 3.Bibliographic sources citation. 

All bibliographic sources have been checked and renumbered.

To 4.Bibliographic source cited twice

Has been corrected.

To 5. Conclusions. 

Conclusion sector has been checked and small adaptations have been done.

To 6.Author Contributions, Data Availability Statement, Acknowledgments, Conflicts of Interest

It has been clarified and completed.  

To 7. References

References chapter has been revised.